# Cutaneous and Vascular Deposits of 18F-NaF by PET/CT in the Follow-Up of Patients with Pseudoxanthoma Elasticum

**DOI:** 10.3390/jcm10122588

**Published:** 2021-06-11

**Authors:** Eugenia Lillo, Antonio Gutierrez-Cardo, Belén Murcia-Casas, Juan Luis Carrillo-Linares, Francisco Garcia-Argüello, Reinaldo Chicharo de Freitas, Isabel Baquero-Aranda, Pedro Valdivielso, María García-Fernández, Miguel Ángel Sánchez-Chaparro

**Affiliations:** 1Molecular Imaging Unit, Centro de Investigaciones Médico Sanitarias (CIMES), Fundación General de la Universidad de Málaga, 29010 Málaga, Spain; elillo@fguma.es (E.L.); jreichicharo@fguma.es (R.C.d.F.); 2Nuclear Medicine Department, Regional Hospital, 29010 Malaga, Spain; algutierrezc@icloud.com; 3Biomedical Research Institute of Malaga (IBIMA), 29010 Malaga, Spain; julucarli@gmail.com (J.L.C.-L.); sfgarcia@fguma.es (F.G.-A.); igf@uma.es (M.G.-F.); masch@uma.es (M.Á.S.-C.); 4Internal Medicine Unit, Virgen de la Victoria Hospital, 29010 Malaga, Spain; belenmurciacasas9@outlook.es; 5Ophtalmology Unit, Virgen de la Victoria Hospital, 29010 Malaga, Spain; imba5@hotmail.com; 6Department of Medicine and Dermatology, University of Malaga, 29010 Malaga, Spain; 7Department of Human Physiology, University of Malaga, 29010 Malaga, Spain

**Keywords:** pseudoxanthoma elasticum, inorganic pyrophosphate, microcalcification, 18F-NaF, PET/CT

## Abstract

Active microcalcification of elastic fibers is a hallmark of pseudoxanthoma elasticum and it can be measured with the assessment of deposition of 18F-NaF using a PET/CT scan at the skin and vascular levels. It is not known whether this deposition changes over time in absence of specific therapy. We repeated in two years a PET/CT scan using 18F-NaF as a radiopharmaceutical in patients with the disease and compared the deposition at skin and vessel. Furthermore, calcium score values at the vessel wall were also assessed. Main results indicate in the vessel walls that calcification progressed in each patient; by contrast, the active microcalcification, measured and target-to-background ratio showed reduced active deposition. By contrast, at skin levels (neck and axillae) the uptake of the pharmaceutical remains unchanged. In conclusion, because calcification in the arterial wall is not specific for pseudoxanthoma elasticum condition, the measurement of the deposition of 18F-NaF in the neck might be potentially used as a surrogate marker in future trials for the disease.

## 1. Introduction

PXE is an autosomal recessive disease caused by the presence of two pathogenic mutations in the ABCC6 gene, which encodes a membrane transporter known as MRP6. This gene is expressed mainly on the cell surface of hepatocytes and in renal tubular cells. Current knowledge on the pathogenesis of the disease indicates that the absence of ABCC6 expression reduces the availability of ATP, a precursor of inorganic pyrophosphate (PPi) when the enzyme ENPP1 [1,2] acts on it. Our group has confirmed that patients with PXE patients have 30% less plasma PPi when compared with age- and gender-matched controls [3].

Patients with PXE suffer from dystrophic calcification from birth that mainly affects elastic fibers. This leads to skin lesions in the flexural areas (neck, axillae, groin), in the artery walls (especially in the lower extremities), and in Bruch’s membrane in the retina [4,5]. Calcification of Bruch’s membrane occurs at the retinal level. In the first decades of life, this becomes visible with the appearance of peau d’orange and angioid streaks (which are diagnostic although asymptomatic), and, in more advanced stages, with the presence of choroidal neovascularization (CNV) in the macular area, which leads to the loss of central vision [6].

Active deposition of hydroxyapatite microcrystals in the vascular wall and skin of PXE patients can be quantified by PET/CT using 18F-sodium fluoride (18F-NaF) as a radiopharmaceutical [7]. Our group has reported that in PXE patients this radiopharmaceutical is also deposited in the flexural areas of the skin (especially the neck and axillae), and that this deposit relates well to the severity of the lesions observed in physical examinations [8]. Identical data have been published more recently [9].

PXE has no specific treatment, although the use of PPi analogs such as etidronate bisphosphonate has successfully reduced the progression of arterial calcification in PXE patients without affecting the deposition of 18F-NaF in the arterial wall [10,11].

This study shows the changes in the deposition of 18F-NaF in the skin and vascular wall, together with arterial calcification measured by CT in PXE patients who underwent a double PET/CT examination carried out two years apart.

## 2. Materials and Methods

Of the 18 patients recruited three years ago, 14 underwent a second 18F-NaF PET/CT examination two years later. Three revoked their consent and one patient was withdrawn due to a serious intercurrent illness. Data included in the analyses were only those 14 who completed both studies. All patients met the diagnostic criteria for the disease [12]. None of the patients was receiving bisphosphonate treatment.

The PET/CT protocols were carried out as published [8]. In summary, 18F-fluoride was produced with a GE PET trace cyclotron and the radiopharmaceutical was prepared using the Tracerlab FXFN synthesis module (GE Healthcare, Chicago, IL, USA). An anion-exchange cartridge (Light Accell QMA, Waters Corporation, Milford, MA, USA) was used to trap 18F-fluoride and radiochemical impurities were removed by washing the cartridge with 10 mL of sterile water for injection. The 18F-NaF activity was then recovered from the QMA with 6 mL of sterile 0.9% saline, and the bulk radiopharmaceutical solution was finally passed through a 0.22 µm filter (Millex GS, Merck Millipore, Darmstadt, Germany) into a sterile empty vial under aseptic conditions. All radiopharmaceutical quality control essays were performed as described [13].

PET/CT acquisition images took place at the Molecular Unit of the “Centro de Investigaciones Médico-Sanitarias” (CIMES), General Foundation of the University of Malaga, Malaga University, on a PET/CT General Electric Discovery ST. All participants received 3.7 MBq/kg of 18F-NaF (maximal dose 370 MBq) and images were taken 90 min after intravenous injection. A low dose CT for attenuation correction was also recorded to evaluate calcification. The uptake of 18F-NaF and calcium deposits in vascular territories of both sides (carotids, ascending, arch, descending and abdominal aorta, iliac, femoral, and popliteal) was measured. The uptake of 18F-NaF was also measured in skin folds on both sides (neck, axillae, elbow, groin, and popliteal) and, for comparison purposes, in the lumbar skin, as an area with no affectation of PXE. From each territory, maximal and mean standard uptake values (SUVs) were recorded; for vascular territories, we adjusted the maximal and mean SUV by the mean values of three measurements at the level of the right auricle [14]. The final results are shown as the target-to-background ratio (TBR, maximal and mean). Vascular calcification was evaluated on a platform Carestream VUE Motion (CareStream Health, Inc., Rochester, NY, USA) software; data are shown as calcium score [15], volume, and mass.

Statistical analyses were carried out in SPSS 26.0 (IBM). Parameters are shown as number (%), mean ± SD, or median (IQ range). The Wilcoxon signed-rank test was used to analyze changes over time. We considered significant a *p* value < 0.05.

Ethical aspects: the study was approved by the Malaga Research Ethics Committee on 15 November 2015. All patients signed the informed consent form.

## 3. Results

### 3.1. Vascular Calcium Parameters on CT

Table 1 shows the mean calcium score values of the 12 vascular territories analyzed using Agatston units. On an individual level (Figure 1), we did not observe regression in any patient, and as a group, progression could be demonstrated in patients (*p* < 0.05). 

### 3.2. Arterial Wall Uptake of 18F-NaF

The values of radiopharmaceutical deposition in the arterial wall detailed in Table 2 show significant changes; thus, in the PET carried out at two years, a significant decrease in the TBR max and TBR mean values is observed.

### 3.3. Skin Uptake of 18F-NaF

Table 3 shows the 18F-NaF deposition values in the skin. The maximum (SUVmax) and mean (SUVmedia) uptake values in the 10 analyzed territories showed no significant variations. In all cases, the uptake values in the skin territories affected by PXE were far higher than the uptake of the radiopharmaceutical in the lumbar skin. Individual skin uptake of 18F-NaF is shown in Figure 2 and Figure 3.

## 4. Discussion

Our data confirm that without disease-specific treatment, calcification progresses in all the arterial territories studied. These data are in line with those obtained by the placebo group of Kranenburg et al. [10,11], in which progression of calcification was observed at one year in the placebo group in contrast to the nonprogression in the group treated with etidronate. Subsequent analysis showed that bisphosphonate treatment had no benefit in terms of ocular disease progression [16].

While vascular wall calcification is common in the general population and is especially related to advanced age, cardiovascular risk factors, and arteriosclerosis [17], fragmentation and calcification of the elastic fibers of the skin and 18F-NaF deposition in the skin is characteristic of PXE. In the GOCAPXE trial, 18F-NaF deposition in the skin of the neck and axillae was higher than that in the lumbar skin, which is considered an area not affected by PXE [9]. For this reason, we believe it is more important to verify that, over two years, this deposition in the skin, especially in the neck and axillae, has remained unchanged. This is consistent with the fact that this is a chronic and progressive disease with no spontaneous regression in affected tissues.

In our study, in the second year, the degree of vascular calcification had increased, despite decreasing the deposition of 18F-NaF in the vascular wall when shown as TBRmax. This apparent contradiction can be explained by the fact that, in already calcified areas, the deposition of the radiopharmaceutical occurs at the periphery of the calcification; thus, the two parameters are not equivalent [7]. A similar finding was described in the Kranenburg study, where the group of patients undergoing etidronate treatment experienced a slowing down of the progression of calcification in the femoral artery without affecting the deposition of the radiopharmaceutical [10].

Our results may be of value when evaluating, in a short space of time, the benefits of new treatments in patients with PXE. A drug that acts by selectively inhibiting the enzyme TNAP [18,19,20] is in phase I of clinical trials and is beginning phase II; the activity of this enzyme is elevated in PXE patients and may be responsible, at least in part, for the reduced levels of PPi [3,21,22]. We speculate that it may be possible to verify the efficacy of this drug in reducing active microcalcification in the skin (and perhaps in the retina) after one or two months of treatment and using 18F-NaF PET/CT; otherwise, this type of treatment could take years before showing benefits.

## 5. Conclusions

Our study confirms that patients who do not receive bisphosphonate treatment show similar SUVmax values in the skin and that this parameter could be used as a surrogate outcome in clinical trials with selective TNAP inhibitors.

## Figures and Tables

**Figure 1 jcm-10-02588-f001:**
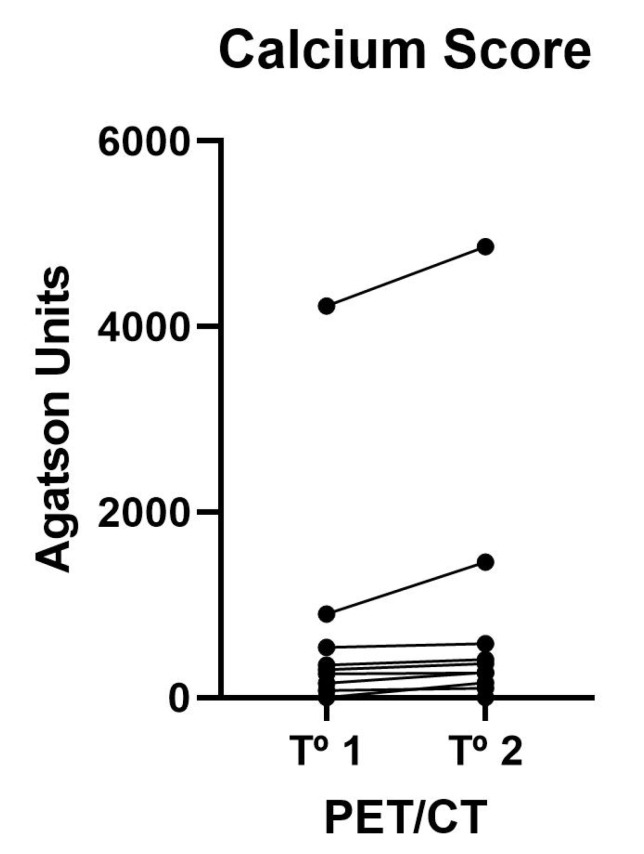
Changes over time for each patient on averaged calcium deposits in vascular territories of both sides (carotids, ascending, arch, descending and abdominal aorta, iliac, femoral, and popliteal).

**Figure 2 jcm-10-02588-f002:**
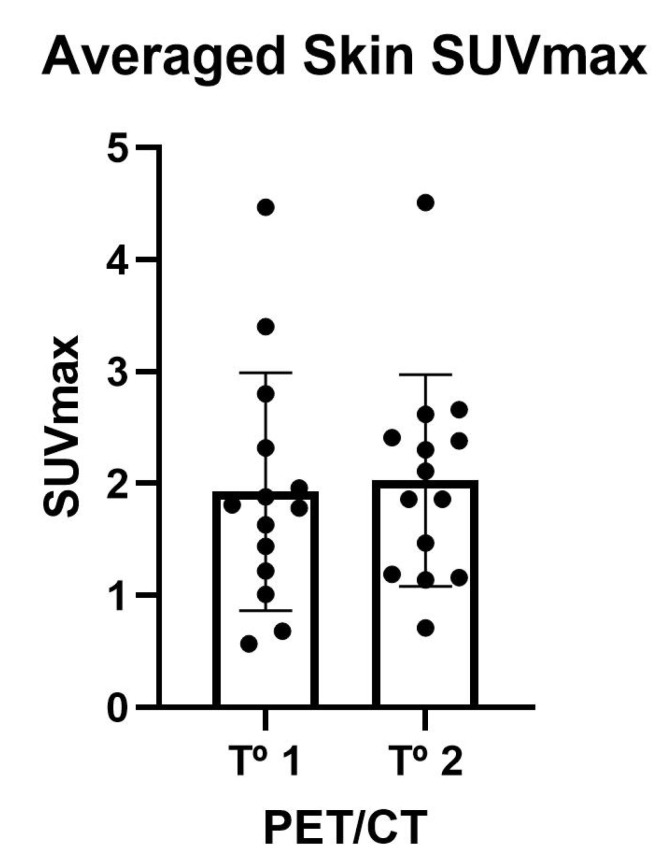
Uptake of 18F-NaF at skin folds on both sides (neck, axillae, elbow, groin, and popliteal area). Figure shows individual and averaged SUVmax of the skin.

**Figure 3 jcm-10-02588-f003:**
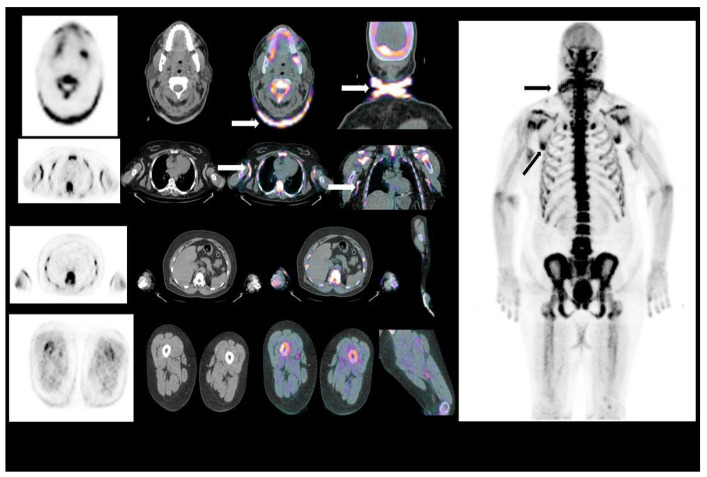
The 18-F-Na uptake in diseased skin at the level of the neck and axillae (arrows) on PET/CT.

**Table 1 jcm-10-02588-t001:** Vascular calcium parameters on CT.

	First	Second	*p* Value
Calcium Score	123 (5–672)	173 (4–518)	0.008
Calcium Volume	169 (7–465)	297 (3–495)	0.023
Calcium Mass	32 (1–95)	57 (1–108)	0.009

Data are shown as median (Q1–Q3).

**Table 2 jcm-10-02588-t002:** TBR max and TBR mean in vascular territories.

	First	Second	*p* Value
SUVMax	0.88 (0.72–0.93)	0.86 (0.76–0.97)	NS
SUVMean	0.67 (0.51–0.74)	0.66 (0.58–0.76)	NS
TBRmax	2.58 (2.01–2.80)	1.94 (1.72–2.25)	0.04
TBRmean	2.10 (1.52–2.34)	1.71 (1.52–1.85)	0.04

Data are shown as median (Q1–Q3).

**Table 3 jcm-10-02588-t003:** Averaged 18F-NaF deposits on skin.

	First	Second	*p* Value
SUVMax	1.79 (1.17–2.44)	1.98 (1.18–2.45)	NS
SUVMean	1.25 (0.90–1.69)	1.12 (0.70–1.40)	NS
LSMax	0.90 (0.64–1.06)	0.84 (0.70–0.95)	NS
LSMean	0.59 (0.44–0.70)	0.62 (0.54–0.67)	NS

Data are shown as median (Q1–Q3); SUVMax = averaged SUVMax on skin territories; SUVMax = averaged SUVMax on skin territories; LS = lumbar skin.

## Data Availability

Data of the study are available upon request.

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
