# Peer review of "Cutaneous and Vascular Deposits of 18F-NaF by PET/CT in the Follow-Up of Patients with Pseudoxanthoma Elasticum"

_jcm, 2021, doi:10.3390/jcm10122588_

Round 1

Reviewer 1 Report

The manuscript ID: JCM-1202887, entitled Cutaneous and vascular deposits of 18F-NaF by PET/CT of the skin in the follow-up of patients with Psudoxanthoma Elasticum is presenting results of a comparative study of 18F-NaF PET acquisitions performed at 2-year intervals in the same PXE patient population (n=14). 

This study offers an interesting message for the future evaluation of PXE therapies in PET, but in our opinion the conclusions are not sufficiently supported by the results.

Regarding the formal aspects of the document, a number of corrections need to be made, for example:

  • The title of the article is not correct (18F-NaF instead of 18-NaF) and not understandable ("cutaneous and vascular deposits of 18F-NaF by PET/CT of the skin....")
  • Captions of the figures are not correct (PET/CT and not PET/TC...), description is too succinct (fig 1, 2...)
  • "is not" instead of "in not" in the introduction ...and many other examples.

A more serious proofreading of the document should have been done before submission.

Concerning the scientific aspects of the manuscript, this work is very interesting but some information is missing and may call into question the authors' conclusions and message.

Here are some questions and comments (not by order of importance) :

Methodology :

1) it is indicated that 4 patients did not underwent a second PET/CT (3+1,  we supposed but it is not clearly indicated that only 14 patients were considered in the results for the PET number 1, could you confirm, please ?

2) As the aim of this work is to carry out a comparative study in PET of the vascular and skin territories of the same population at 2 years of difference, in our opinion it would have been essential to have the clinical information of the patients at time 1 and at time 2 (phenodex score of the patients, more detailed information concerning evolution of the skin and vascular territories...). It is hazardous to draw conclusions for the reader only based on PET without information concerning evolution of the pathology in the studied areas. This is a key point in our opinion.

Results-Discussion

3) Presentation of the results is not homogenous, while calcium score is presented as the individual changes in patient, PET/CT is presented as the population changes ...

4) The message presented by the authors is very interesting, i.e. that skin lesions (rather than vascular territories) are the best target for 18F-NaF PET/CT evaluation of new PXE therapies. However, the way this conclusion is reached is a bit inadequate in our opinion. 

Indeed, the results presented are limited to saying that the TBR index at the vascular level decrease between the 2 PET scans, which would disqualify the vascular study, whereas the study of skin lesions is relevant because the SUVs are stable between PET 1 and PET 2. In our opinion, this reasoning is a bit flawed because important points are omitted and not discussed by the authors, you will find some of them below...

  • Between the 2 TEP, the SUV in the vascular territories is as stable than the SUV measurement in skin lesions, why in one case (skin) it is a good evaluation tool and not in the other (vascular)?
  • If an equivalent of TBR is used to evaluate calcification in skin lesions what is the guarantee not to obtain the same results that in vascular territories ?
  • Concerning skin lesions, how can we know that it is normal or relevant to observe no  SUV variation between the two TEP knowing that as a reader we do not have clinical information on these lesions in the study population (cf. remark 2)

Author Response

The manuscript ID: JCM-1202887, entitled Cutaneous and vascular deposits of 18F-NaF by PET/CT of the skin in the follow-up of patients with Psudoxanthoma Elasticum is presenting results of a comparative study of 18F-NaF PET acquisitions performed at 2-year intervals in the same PXE patient population (n=14). 

This study offers an interesting message for the future evaluation of PXE therapies in PET, but in our opinion the conclusions are not sufficiently supported by the results.

Regarding the formal aspects of the document, a number of corrections need to be made, for example:

  • The title of the article is not correct (18F-NaF instead of 18-NaF) and not understandable ("cutaneous and vascular deposits of 18F-NaF by PET/CT of the skin....") Already corrected
  • Captions of the figures are not correct (PET/CT and not PET/TC...), description is too succinct (fig 1, 2...) Figure captions have been modified and being more informative
  • "is not" instead of "in not" in the introduction ...and many other examples.  Text has been reviewed and error amended

A more serious proofreading of the document should have been done before submission. We have carefully reviewed the document and we hope to be improve it final presentation.

Concerning the scientific aspects of the manuscript, this work is very interesting but some information is missing and may call into question the authors' conclusions and message.

Here are some questions and comments (not by order of importance) :

Methodology :

1) it is indicated that 4 patients did not underwent a second PET/CT (3+1,  we supposed but it is not clearly indicated that only 14 patients were considered in the results for the PET number 1, could you confirm, please ? We confirm that issue and have added a short comment on the manuscript indicating that only those 14 pairs of studies were analysed in the current paper.

2) As the aim of this work is to carry out a comparative study in PET of the vascular and skin territories of the same population at 2 years of difference, in our opinion it would have been essential to have the clinical information of the patients at time 1 and at time 2 (phenodex score of the patients, more detailed information concerning evolution of the skin and vascular territories...). It is hazardous to draw conclusions for the reader only based on PET without information concerning evolution of the pathology in the studied areas. This is a key point in our opinion. We did not expect major changes in only two years at the skin levels, so we did not perform a second detailed cutaneous examinations. However,  it was clear, at least in our first paper, that there was an association between severity at the neck level with the deposition on the radiopharmaceutical.

Results-Discussion

3) Presentation of the results is not homogenous, while calcium score is presented as the individual changes in patient, PET/CT is presented as the population changes ...

For calcium score as well the 18F-NaF deposition on the skin and vessel wall we have shown mean individual (se Fig 1 and 2) and averaged data (Tables). In Fig 1, for instance, each black circle represents the mean values of calcium score at the vascular territories explored. The same is true for Fig 2: each black circle is the mean value of skin deposit of the radiopharmaceutical at the skin levels assessed.

4) The message presented by the authors is very interesting, i.e. that skin lesions (rather than vascular territories) are the best target for 18F-NaF PET/CT evaluation of new PXE therapies. However, the way this conclusion is reached is a bit inadequate in our opinion. 

We really appreciate the comment of the reviewer, because it focuses on the main message of our contribution. See below

Indeed, the results presented are limited to saying that the TBR index at the vascular level decrease between the 2 PET scans, which would disqualify the vascular study, whereas the study of skin lesions is relevant because the SUVs are stable between PET 1 and PET 2. In our opinion, this reasoning is a bit flawed because important points are omitted and not discussed by the authors, you will find some of them below...

  • Between the 2 TEP, the SUV in the vascular territories is as stable than the SUV measurement in skin lesions, why in one case (skin) it is a good evaluation tool and not in the other (vascular)?

Deposition of 18F-NaF in arterial wall is a very well-studied phenomena and included in many arteriosclerosis research. Calcification and active microcrystal deposition detected by PET/CT is “physiological” in arteries and enhanced by systemic disorders. This is not the case for PXE patients and for cutaneous deposition. As it has been published, there is no deposition of 18F-NaF in the skin (Nuclear Medicine Communications 2017, 38:810–819). 

Natural evolution of calcification in arteriosclerosis leads to a stabilization of calcium deposits that cause decline in the necessary mechanism to 18F-NaF deposition: free terminal of hydroxiapatite. 18F ions exchange with hydroxyl ions (OH–) on the surface of the hydroxyapatite to form fluoroapatite. This exchange occurs at a rapid rate in newer bone or calcified tissue; however, the actual incorporation of 18F ions into the crystalline matrix of bone may take days or weeks.

. Radiographics, 34(5), 1295–1316. https://doi.org/10.1148/rg.345130061

. J Nucl Med 2010;51(12):1826–1829.

. J Nucl Med 1969;10(1):8–17.

  • If an equivalent of TBR is used to evaluate calcification in skin lesions what is the guarantee not to obtain the same results that in vascular territories?

Obviously, there is no guarantee. TBR is a well-stablished parameter in the study of vascular wall inflammation using 18F-FDG and 18F-NaF and according the guidelines takes the vascular blood pool at venous cava or right auricle as denominator. The problem using TBR for skin evaluation is that has not been verified previously. In our first paper, we took SUVmax at the lumbar skin for the denominator, because clinically is not clearly affected by PXE changes. That approach was strongly put in question by one of the referees, and she/he suggested to avoid it.

  • Concerning skin lesions, how can we know that it is normal or relevant to observe no  SUV variation between the two TEP knowing that as a reader we do not have clinical information on these lesions in the study population (cf. remark 2).

PXE disease is a chronic and progressive disease where the structural changes at the retina, skin or vessel wall never show regression. Our finding that at two years the deposition of the radiotracer remains active and unchanged over time let us to speculate with the value of the PET/CT in the design of new drugs, as mentioned in the discussion.

To clarify the point raised by the reviewer, we have added   short comment on that issue in the discussion.

Reviewer 2 Report

The paper by Lillo et al. describes the use of 18F-NaF-PET-CT as a method to detect and follow-up vascular and skin calcifications in PXE patients They show that calcification of the arteries increases after a two year follow-up while calcification of the skin does not change.

The findings regarding the arteries are not entirely new, similar results have been published by Kranenburg et al. after a one year follow-up.

Unfortunately, the paper is not very well written, especially the abstract has to be rewritten because of numerous errors, e.g. “whether” instead of “wether”, “”the deposition at skin and vessel wall as well the vessel wall”, “because calcification in arterial wall in not specific”.

There are additional spelling errors and errors in grammar throughout the manuscript, which make it difficult to understand the content of the paper. In the method section, it should be stated that the follow-up scan was performed two years after the first scan. The meaning of a decrease in TBR max and TBR mean values should be explained.  It only can be guessed that this translates to an increase of calcification.

Author Response

The paper by Lillo et al. describes the use of 18F-NaF-PET-CT as a method to detect and follow-up vascular and skin calcifications in PXE patients They show that calcification of the arteries increases after a two year follow-up while calcification of the skin does not change.

The findings regarding the arteries are not entirely new, similar results have been published by Kranenburg et al. after a one year follow-up.

Unfortunately, the paper is not very well written, especially the abstract has to be rewritten because of numerous errors, e.g. “whether” instead of “wether”, “”the deposition at skin and vessel wall as well the vessel wall”, “because calcification in arterial wall in not specific”.  

We appreciate the comment of the reviewer; the document has been revised and errors have been amended. Changes in the text have been emphasized in red.

There are additional spelling errors and errors in grammar throughout the manuscript, which make it difficult to understand the content of the paper. In the method section, it should be stated that the follow-up scan was performed two years after the first scan.

Done

The meaning of a decrease in TBR max and TBR mean values should be explained.  It only can be guessed that this translates to an increase of calcification.

We really agree with the referee. TBR max and TBR mean (which indicate active deposition of calcium) are dissociate from the chronic, stable, deposition of calcium at the arterial wall. We offered one explanation for this apparent controversy (paragraph 144-151 in discussion). In fact, in the paper from Kranenburg the  progression of calcification at femoral artery was halted one year after therapy with etidronate, but there were no changes at TBR max.

Round 2

Reviewer 1 Report

We still find typing errors previously reported (PET/TC instead of PET/CT on the figures!!!).

The response of the authors and the few modifications made allow some improvements to the manuscript, but the answer and the explanations given by the authors, in particular concerning the absence of clinical observation of the lesions at 2 years after the first PET scan, are not convincing to justify and conclude to a real value of the SUV measurement (cutaneous territories), to carry out the follow-up of this pathology and to evaluate a therapy.

I repeat, the authors' message is attractive but the absence of change in SUV is a descriptive fact and cannot be scientifically justified because we do not know the clinical status of the patients neither at time 1 nor at time 2.  So how can we conclude that the absence of change in SUV at the skin level is

  1. Normal and expected
  2. a surrogate markers of outcome in clinical trials

We agree with the authors that PXE lesions do not regress over time, so at a minimum we must have a maintenance of SUV over time (between time 1 and 2). But as we do not know the clinical status of the skin lesions, a progression of the latter, even if slow, cannot be excluded. In which case this progression should have resulted in an increase in SUV. 

So what can we conclude from this study in the face of an SUV that did not vary? 
1) it is a good marker because at the same time the lesions have not progressed
2) it is a bad marker because at the same time the lesions have progressed.

The arguments and conclusions of the authors seem too speculative to be published as is. 

Author Response

We really appreciate the comments by the reviewer, because they are focused on the core of our investigation.

The clinical status was well known in the participants in the study. In fact, in our first report with the initial findings, were showed detailed clinical data on PXE especially at the skin folds (Clin. Med. 2020, 9, 1393, doi:10.3390/jcm9051393).

Clinical staging in PXE is quite difficult especially using the Phenodex Score (¡which is the best Score available!). Skin abnormalities are ranked from 0 to 4 and very observer-dependent. It is clear to note that at any specific skinfold that  a progression has been made from stage 3 to 4,  but not to see changes in the same stage 3 (the same for others stages). We are absolutely convinced that in two years the skin lesions must be worsened but not enoguh to see clinically evident for the dermatologist and to change from one to another stage. That was the reason we don't include a re-staging of skin lesion at the second PET/CT, made only two years later. On the other hand, the progression of the skin lesion indicates that more elastic fibers has been already calcified (the final stage of calcification), but the deposition of 18F-NaF indicates active microcalcifcation  (an early stage in the calcification that is clinically silent).

Which is also well known, is that PXE clinical trials are limited because you must follow-up skin, vascular o retinal abnormalities for many years just to know if there is any benefit in terms of efficacy but also in safety. So we are forced to look for intermediate surrogate end-points that can anticipate benefits in the medium or long-term.

One of this intermediate end-point might be changes in serum PPi levels (NCT04660461) or the deposition of calcium at the vessels or the skin. Other might be the deposition of 18F-NaF at the skin-folds. If you take the works of the Krannenbrug's groups, etidronate was able to show delay in the calcification of the vessel wall but not any benefit in the retina. They did not look at the skin.

We believe that our findings of the unchanged deposition of 18F-NaF at two-years in the skin of not reated patients with PXE are relevant and independent of the potential skin changes visible for an experienced dermatologist and of course my be a surrogate end-point for a short-time trial. Active microcalcification is pathogenic of the disease; if a drug (TNAP inhibitor or simmilar) is able to reduce in several months the deposition of the isotope, then is clear that it might be delay the progression of the disease.

PET/CT have been added to figures.

Reviewer 2 Report

In the abstract, it has to read:  "...the uptake of the pharamceutical remains unchanged".

Author Response

We have amended the abstract following  the sugestion of the reviewer.